# Drug Evaluation Based on a Multi-Channel Cell Chip with a Horizontal Co-Culture

**DOI:** 10.3390/ijms22136997

**Published:** 2021-06-29

**Authors:** Gyeong-Ji Kim, Kwon-Jai Lee, Jeong-Woo Choi, Jeung Hee An

**Affiliations:** 1Department of Food and Nutrition, KC University, Seoul 07661, Korea; kgj8495@hanmail.net; 2Department of Biomedical Engineering, Sogang University, Seoul 04107, Korea; 3College of H-LAC, Daejeon University, Daejeon 34520, Korea; kjlee@dju.kr; 4Department of Chemical and Biomolecular Engineering, Sogang University, Seoul 04107, Korea

**Keywords:** multi-channel, cell chip, 3D culture, drug screening, toxicity evaluation

## Abstract

We developed a multi-channel cell chip containing a three-dimensional (3D) scaffold for horizontal co-culture and drug toxicity screening in multi-organ culture (human glioblastoma, cervical cancer, normal liver cells, and normal lung cells). The polydimethylsiloxane (PDMS) multi-channel cell chip (PMCCC) was based on fused deposition modeling (FDM) technology. The architecture of the PMCCC was an open-type cell chip and did not require a pump or syringe. We investigated cell proliferation and cytotoxicity by conducting 3-(4,5-dimethylthiazol-2-yl)-2,5-dphenyltetrazolium bromide (MTT) and lactate dehydrogenase (LDH) assays and analysis of oleanolic acid (OA)-treated multi-channel cell chips. The results of the MTT and LDH assays showed that OA treatment in the multi-channel cell chip of four cell lines enhanced chemoresistance of cells compared with that in the 2D culture. Furthermore, we demonstrated the feasibility of the application of our multi-channel cell chip in various analysis methods through Annexin V-fluorescein isothiocyanate/propidium iodide staining, which is not used for conventional cell chips. Taken together, the results demonstrated that the PMCCC may be used as a new 3D platform because it enables simultaneous drug screening in multiple cells by single point injection and allows analysis of various biological processes.

## 1. Introduction

Techniques for three-dimensional (3D) scaffold fabrication include solvent casting, gas forming, salt leaching, fiber binding, and membrane lamination [1]. However, it is difficult to control the shape and pore size of interconnected porous structures and scaffolds [2,3,4]. Generally, a 3D organ printing system can overcome the limitations of existing methods by distributing the filaments of the material in a layer-by-layer manner to produce scaffolds with various sizes, shapes, and internal architectures [5]. In addition, the porosity of a 3D printed scaffold allows the transport of nutrients and metabolic waste, making these scaffolds suitable surfaces ideal for cell adhesion and proliferation [6,7]. However, a single 3D cell culture platform cultured with one type of cell is insufficient to represent the complex characteristics of the human body [8,9]. Moreover, a single 3D cell culture platform is limited to expressing drug interactions in the human body [9]. Therefore, a multi-channel integrated 3D single cell culture platform can provide the ability to control the dynamic elements of the microenvironment and support the introduction of the mechanical and chemical signals missing from a single platform.

Various approaches have been used to fabricate platforms that integrate multiple organs. For example, Zhang et al. fabricated a 3D microfluidic cell culture system for culturing four different human cell types to mimic four human organs: liver, lung, kidney, and fat [10]. Rajan et al. fabricated an in vitro multi organoid system (liver, cardiac, lung, endothelium, brain, and testes) that enabled the parallel assessment of drug efficiency and toxicity on multiple 3D tissue organoids [8]. In another study, Novak et al. developed an automated platform involving vascularized, two-channel organ chips (intestine, liver, kidney, heart, lung, skin, blood–brain barrier, and brain) [11]. However, these platforms have certain disadvantages. First, most multi-organ platform technologies require a complex external pump connected to the channel of the device through a tube to perfuse the culture medium within the device [12,13]. Second, microfluidic devices have closed microchannel passages, which makes cell seeding, harvesting, staining, and analysis difficult [14]. It is also difficult to insert a pipette tip into the microfluidic inlet while precisely controlling the placement of the cell suspension for a given culture area [14]. Therefore, in this study, we developed a simple, pumpless and open cell chip to address these disadvantages. In addition, human cancer cell lines (neuroblastoma and cervical cancer cells) and normal cells (liver and lung cells) were cultured on cell chips to observe anticancer effects and cytotoxicity in various organs when treated with an anticancer drug.

We developed a 3D cell platform using 3D printing technology. We named this new 3D cell platform “polydimethylsiloxane (PDMS) multi-channel cell chip”. We fabricated the PDMS multi-channel cell chip (PMCCC) containing a 3D scaffold using a fused deposition modeling (FDM) technology. The 3D scaffolds were consisting of FDM technology, fabricated with a polycaprolactone (PCL) material. The PMCCC was used to analyze the proliferation and toxicity of human cancer cell lines and normal cells by 3-(4,5-dimethylthiazol-2-yl)-2,5-diphenyl tetrazolium bromide (MTT) assay, lactate dehydrogenase (LDH) assay, and Annexin V-fluorescein isothiocyanate (FITC)/propidium iodide (PI) staining. This is the first study to report the development of a multi-channel cell chip containing a PCL3D scaffold platform that can be easily used to simulate 3D cell growth and detect anticancer effects by simultaneously culturing four cell types in vitro.

## 2. Results and Discussion

### 2.1. The Characteristic of PDMS Multi-Channel Cell Chips

Three-dimensional printing technologies, such as FDM, use various materials to fabricate 3D scaffolds. Figure 1 shows a schematic diagram of the PMCCC. As shown in Figure 2A(b), the size of the cell injection holes of the PMCCC was 12 × 12 × 3 mm, and the culture media volume was 430 µL. The central hole size was 2 × 2 × 5 mm. Four cell injection holes of the PMCCC could be used to culture four different types of cells. After the attachment of cells for 4 h, the same medium or drug could be supplied or collected simultaneously for the four cell types through a central hole that acted as an inlet and outlet in the center. In addition, when the medium or drug was injected through the inlet (central hole), excess liquid was released through the cell injection or harvest hole. Therefore, only a certain amount of medium or drug could be present in the cell culture chamber with a square shape. Inside the chamber, there was a PCL 3D scaffold that allowed cells to grow and achieve a 3D morphology. Additionally, the PMCCC allowed simultaneous observation of the effects of one drug on four different cell types.

We observed the possibility of culture medium migration in a multi-channel cell chip. As shown in Figure 2A(c), the channel width of the PMCCC was 200 µm. The PMCCC showed simultaneous migration of the culture medium. Generally, the width of a channel directly affects fluid flow [15]. To maintain laminar flow within the chip, the channels should be as small as required to maintain the functionality of the chip [15]. The diameter of a cell ranges from 5 μm to 50 μm [15]. For cells to have a good cell–cell interaction, multiple cells must be present together [15]. This means that the minimum channel width should allow at least four cells to attach perpendicularly (side by side) to the fluid flow [15]. Hence, the minimum width should be 200 μm [16]. Goh et al. suggested an affordable tool for the fabrication of microchannels measuring 200 µm in width and 100 µm in height using polyvinyl alcohol, and demonstrated that they are compatible with a wide range of polymeric matrixes [17]. Saggiomo et al. used an FDM technique to extrude an inexpensive and commercially available acrylonitrile butadiene styrene (ABS) scaffold using a 500 µm diameter nozzle. The scaffold was then cured with PDMS and acetone was added to dissolve the ABS scaffold to fabricate a microfluidic device [18]. Our PMCCC was fabricated using industrial polylactic acid (PLA) filaments, with a channel width of 200 µm. As shown in Figure 2A(d), the cell culture chamber size of the PMCCC and silicon wafer multi-channel cell chip were 12 × 12 × 3 mm and 5 × 5 × 0.5 mm, respectively. The PMCCC could hold media volumes of 430 µL. Evaporation can negatively affect the cell culture process because of high osmotic pressure and concentrating effects [19,20,21]. Our PMCCC had a low effect from evaporation. Therefore, the PMCCC was suitable for the cell culture.

### 2.2. Characteristics of the PCL 3D Scaffolds Fabricated Using FDM

The surfaces of the PCL 3D scaffolds, fabricated using the FDM technique, were analyzed by SEM (Figure 2B). Figure 2B(a–d) shows the high-magnification SEM image of an enlarged area of the PCL 3D scaffold. The dimensions of the PCL 3D scaffold were 10 × 10 × 0.75 mm. The filament diameter and pore size of the PCL 3D scaffolds were 101.25 and 166.75 µm, respectively. The FDM technique has the advantage of being able to fabricate scaffolds of various sizes, shapes, and materials. In addition, the surface area of the PCL 3D scaffold fabricated is much larger than a conventional cell culture dish. Scaffolds can accommodate a high density of cells by providing a high surface-to-ratio structure, using a porous structure with interconnected pores and micropillar [22]. The porosity of scaffolds provides space for cell penetration and growth and enables advantageous mass transport properties [22].

### 2.3. Morphology of Cells Cultured in the PDMS Multi-Channel Cell Chip

Figure 3A shows the optical images of the four cell types (BNL-CL2, L-132, U87, and HeLa) cultured in the PMCCC. Generally, attached cells cultured in 2D display a flat morphology and random cell spreading. We observed that the cells cultured on the PMCCC grew flat on the bottom and the pillars. In addition, cell adhesion stability was observed through optical images every 2 h after cell culture. After 6 h, each of the four cell types did not mix, and the cells attached firmly to the surface of the dish and scaffold. However, the SEM images showed that the cell morphology was different from that of the optical images. The SEM image in Figure 3B shows HeLa cells cultured on the PMCCC. Compared to 2D plates, the large surface area of the PCL 3D scaffold allowed more cells to attach. The plain view image showed that the HeLa cells were cultured in the PCL 3D scaffold (Figure 3B(a)). High-magnification images showed the 1st floor and 2nd floor HeLa cells (Figure 3B(b)) and enlarged images of the HeLa cells (Figure 3B(c,d)). As shown in Figure 3B(d), the cells on the surface of the PCL material showed partial cell aggregation and spheroid formation. Our results showed that the HeLa cells cultured on the PCL 3D scaffold maintained high levels of cell viability and possessed excellent cellular morphology. Cells grown on 3D scaffolds were able to form aggregates and spheroids, allowing increased cell–cell contact, when compared with the 2D system [23]. PCL is a commonly used polymer for the 3D printing of scaffolds for bone tissue [24]. Mesenchymal stem cells are incorporated with PCL-based scaffolds and have been reported to improve bone regeneration when applied to a rabbit femur [6]. The use of these materials for 3D printing allows the fabrication of scaffolds with highly reproducible and uniform open pore sizes [25]. We confirmed the application of the PCL material for 3D culture of normal cells (lung and liver) and cancer cells (glioblastoma cell and cervical carcinoma) in a PMCCC.

### 2.4. The Comparison of Cell Morphology of the PDMS Multi-Channel Cell Chips and 2D Cell Culture

In Figure 4, we compared the cell morphology in the same area of the 2D plate and PMCCC. The HeLa cell culture on the 2D plate showed a flat morphology in Figure 4A. Whereas, seeded cells on the PCL 3D scaffolds were infiltrated, attached and proliferated inside the scaffolds Figure 4B. Additionally, HeLa cells cultured on the PMCCC were partially flat or displayed an aggregated morphology. Furthermore, the surface of cells cultured in the PMCCC secreted more extracellular matrix compared with the surface of 2D-cultured cells. In addition, Figure 4C shows the number of aggregation cells in the PMCCC and 2D cell culture. Three aggregated cells were observed on the PMCCC. However, 2D cell culture did not show aggregation cells. The interconnected pores of 3D scaffolds facilitate the transport of biomolecules and cells [26]. Therefore, we performed drug screening on the PMCCC combined with PCL 3D scaffolds compare to 2D-cultured cells. Additionally, we focused on whether conventional molecular biology analysis methods, such as MTT LDH assays, and fluorescence-activated cell sorting, can be applied to the PMCCC.

### 2.5. Evaluation of Proliferation of Cells Cultured in PDMS Multi-Channel Cell Chip Treated with Drug

Various biological activities have been reported for oleanolic acid (OA), including chemo-preventive, hepatoprotective, tumor-suppressive, contraceptive, anti-inflammatory, antioxidant, antimicrobial, antiparasitic, antiviral, anticancer, and antineoplastic effects. The tumor suppressive activity of OA was investigated in cancer cell lines involved in glia cell, liver cell, breast cell, prostate cell, cervical and lung cell [27,28,29]. To investigate the effect of OA on the proliferation and biocompatibility of BNL-CL2, L-132, U87, and HeLa cell lines in the PMCCC, we treated cells with 0, 50, 100, 200, and 300 µg/mL OA for 24 h and assessed cell proliferation and viability using the MTT assay (Figure 5). However, the results of the silicon wafer multi-channel cell chip are not shown because the MTT assay could not be performed on a small area. The MTT assay is based on the ability of viable cells to enzymatically convert to other colors, whereas dead cells following toxic damage cannot transform color [30]. These reactions of the MTT assay are mediated by dehydrogenase enzymes related to the mitochondria and endoplasmic reticulum [31,32]. When comparing the potency of OA in 2D and PMCCC at the same seeding density, we observed statistically significant differences. The proliferation of BNL-CL2 cells treated with 300 µg/mL OA in the 2D plate and PMCCC was 41.76% and 107.42%, respectively. In the case of L-132 cells treated with 50 µg/mL OA, the cell viability on the PMCCC decreased by 23.44% compared with the 2D plate-cultured cells (63.30%). However, when L-132 cells were treated with a high concentration (300 µg/mL) of OA, cell proliferation on the 2D plate and the PMCCC was 5.86% and 43.77 %, respectively. In addition, U87 cells treated with 300 µg/mL OA in the 2D plate and the PMCCC showed a cell proliferation of 16.50% and 79.38%, respectively. Treatment of HeLa cells with 50 µg/mL OA in the PMCCC showed cell proliferation of 95.46%, whereas in 2D plates, cell proliferation decreased to 68.66%. Furthermore, the four cell types cultured on the PMCCC showed that the higher the concentration of OA treatment, the higher the drug resistance compared to the 2D-cultured cells. Cells cultured on 3D scaffolds may have increased or decreased drug resistance compared to those cultured on 2D scaffolds [23]. Moreover, depending on the 3D environment and cell–cell and cell–matrix interactions, cells may become resistant or sensitive to certain drug treatments [23]. Nevertheless, the phenomenon of cell spheroids displaying elevated chemoresistance to anticancer reagents has been attributed to several mechanisms, including decreased penetration of anticancer drugs, increased pro-survival signaling, and/or the upregulation of genes conferring drug resistance [32,33]. Drug resistance plays a major role in the failure of certain chemotherapeutic agents in the treatment of solid tumors [34]. Various spheroid structures show drug resistance and are influenced by the expression of different oncogenic molecules [33,34]. Fontoura et al. reported that after treatment with dacarbazine, cells grown in 3D culture models an showed increased resistance compared to cells grown in monolayers [23]. Imamura et al. demonstrated that dense spheroids increased the resistance to paclitaxel and doxorubicin drugs when compared to 2D systems and looser spheroids [35]. Thus, our results suggest that the PMCCC increased chemoresistance of cells to anticancer drugs and could be used for high throughput investigations of the efficacy and toxicity of drugs, gene expression in spheroids, and numerous other cellular and biochemical assays.

### 2.6. Evaluation of Toxicity of Cells Cultured in PDMS Multi-Channel Cell Chip Treated with Drug

Necrotic cell death is assessed by measuring the degree of damage to the plasma membrane and is usually analyzed by the LDH assay [36]. LDH is a soluble cytoplasmic enzyme that is released into the extracellular space when the plasma membrane is damaged [37]. Thus, the LDH assay is based on the detection of enzyme leakage into the culture medium [38]. We analyzed cytotoxicity and drug resistance of the cells cultured on the PMCCC treated with OA (0, 50, 100, 200, and 300 µg/mL) using the LDH assay (Figure 6). The BNL-CL2 cells treated with 50 µg/mL OA and cultured on 2D plates showed 30.11% cytotoxicity, whereas the cells cultured on the PMCCC showed 12.21% cytotoxicity. In the PMCCC, BNL-CL2 cells treated with 200 µg/mL and 300 µg/mL OA were decreased by 18.38% and 16.06%, respectively, compared to those in the 2D plate. Treatment of L-132 cells cultured on the PMCCC with 50 µg/mL OA showed no cytotoxicity. In contrast, 2D-cultured cells treated with 50 µg/mL OA showed 17.25% cytotoxicity. The cytotoxicity of L-132 cells cultured on the PMCCC was significantly reduced when treated with 200 µg/mL (1.75-fold) and 300 µg/mL (2.00-fold) OA, compared to that of 2D-cultured cells. When treated with a high concentration (200 µg/mL and 300 µg/mL) of OA, U87 cells cultured on the PMCCC showed decreased cytotoxicity compared to the 2D plate-cultured cells. In the PMCCC, HeLa cells treated with OA showed significantly increased drug resistance compared to cells cultured on 2D plates. HeLa cells cultured on the PMCCC and treated with 50 µg/mL, 100 µg/mL, 200 µg/mL, and 300 µg/mL OA decreased the cytotoxicity by 5.26, 3.05, 2.47, and 2.02-fold, respectively, when compared with the 2D plate-cultured cells. In conclusion, cells cultured on the PMCCC showed less damage to membrane integrity on OA treatment compared to the 2D-cultured cells.

### 2.7. Apoptosis Effect in PDMS Multi-Channel Cell Chip Treated with Drug

Next, we determined whether the anti-proliferative effects of anticancer drugs, such as OA, are related to apoptosis of cells cultured on the PMCCC, by quantifying apoptosis (Figure 7A). The cells (BNL-CL2, L-132, U87, and HeLa cells) were treated with 50 µg/mL and 100 µg/mL OA for 24 h. Apoptotic cells were quantified by Annexin V-FITC/PI staining (Figure 7B). OA non-treated cells cultured on the PMCCC did not show apoptosis. These results confirmed that the PMCCC was biocompatible. For BNL-CL2 cells treated with 50 µg/mL OA, the relative percentages of viable cells, early apoptotic cells, and late apoptotic cells were 42.8%, 22.8%, and 33.3%, respectively. In the case of BNL-CL2 cells treated with 100 µg/mL OA, the number of late apoptotic cells increased by 61.5%. L-132 cells treated with 50 µg/mL OA did not show apoptotic cells, while those treated with 100 µg/mL OA showed 20.9% necrosis. For U87 cells treated with 100 µg/mL OA, the relative percentages of viable cells, late apoptotic cells, and necrosis were 58.1%, 20.5% and 18.8%, respectively. HeLa cells treated with 0 and 50 µg/mL OA did not show apoptotic cells or necrosis; however, treatment with 100 µg/mL OA resulted in an increase in late apoptotic cells (20.7%) and necrosis (11.65%). Apoptotic cells have been found to be highly immunogenic as well as release various anti-inflammatory molecules [39]. Moreover, because apoptotic cells are removed by phagocytosis, they do not usually progress to secondary necrosis [40,41]. In contrast, the necrosis process is accidental and passive due to environmental disturbances caused by the release of inflammatory cell contents [42]. Additionally, the necrotic cell death is caused by an inflammatory response [36,43]. We confirmed that the effect of the drug on BNL-CL2 cells cultured on the PMCCC occurred via apoptosis mechanisms, whereas L-132 cells showed cell death by necrosis. U87 cells showed the same ratio of apoptosis and necrosis, whereas the cell death mechanism of HeLa cells showed higher apoptosis than that of necrosis. These results showed that the mechanism of cell death differed depending on the cell type. In conclusion, our findings showed that the PMCCC can be used for simultaneous analysis of the mechanism of cell death, such as necrosis or apoptosis, using Annexin V-FITC/PI staining simultaneously in four different types of cells.

### 2.8. Correlation of Drug Resistance through MTT Assay, LDH Assay, and Annexin V-FITC Staining in HeLa Cells Cultured on PDMS Multi-Channel Cell Chip

Figure 8 shows the correlation of the drug effect through MTT and LDH assays, and Annexin V-FITC/PI staining in HeLa cells cultured on the PMCCC. Cell viability of drug resistance was correlated to the MTT and LDH assays, and Annexin V-FITC/PI staining using exponential regression. First, when the correlation between the MTT assay and LDH assay was analyzed, the results showed a positive correlation with R^2^ values of 0.8903. Second, the correlation between the MTT assay and Annexin V staining showed R^2^ values of 0.9876. Finally, the R^2^ values of the LDH assay and annexin V staining were 0.7659. The strongest correlation of HeLa cells cultured in PMCCC was correlated with the MTT assay and Annexin V-FITC/PI staining. When experimenting in a new environment, correlation analysis of various assays is important to find suitable assays. In addition, the experimental consistency can be confirmed through the correlation analysis. Motskin et al. demonstrated that synthetic colloid and gel hydroxyapatite uptake in human monocyte-derived macrophages correlated with each of the cytotoxicity assays (confocal live–dead, MTT, and LDH assays) by exponential regression analysis [44]. Our results suggest that the MTT assay, LDH assay, and Annexin V-FITC/PI staining are suitable for the PMCCC.

## 3. Materials and Methods

### 3.1. Fabrication of a PDMS Multi-Channel Cell Chip with a 3D Scaffold Using a 3D Bioprinter

We showed the fabrication process of PMCCC in Figure 1. A 3D scaffold consisting of a three-floor structure was fabricated by FDM technology using PCL at 85 °C on a cell culture dish (Table 1). A 3D bioprinter (INVIVO; ROKIT, Geumcheon-gu, Seoul, Korea) was set such that the PCL was extruded through a nozzle (diameter: 0.2 mm) and deposited in accordance with predetermined designs. The NewCreatorK program (Version 1.57.70, ROKIT INVIVO Corp., Seoul, Korea) was used to generate the G-code to convert the 3D object files for compatibility with a 3D bioprinter. The printing speed was optimized to a feed rate of 2 mm/s. The sizes of the first, second and third floors were set to 6 × 6 × 0.25 mm, 8 × 8 × 0.25 mm and 10 × 10 × 0.25 mm, respectively. The distance between each of the three floors was set to 0.2 mm. The PCL 3D scaffolds were fabricated four per cell dish. The printing time for the three-floor structure was approximately 2 h. We printed four PCL scaffolds on a 100 mm cell dish as shown in Figure 1A. The morphology of the PCL 3D scaffold was visualized using a scanning electron microscope (SEM) (ZEISS Gemini 2, Oberkochen, Germany).

The multi-channel cell chip was fabricated by a 3D bioprinter, using PLA filaments and PDMS as shown in Figure 1B,C. The PLA filaments were extruded through a nozzle (diameter, 0.2 mm) and deposited in accordance with predetermined designs using the NewCreatorK program on the 60 mm cell culture dish (Table 1). The PLA mold showed in Figure 2A(a). The horizontal and vertical line sizes size was set to 24 × 0.3 × 0.3 mm. The size of the cylinder in the middle (central hole: medium change hole) of the line was 2 × 2 × 5 mm. The size of the square structure to make each cell culture chamber was designed to be 12 × 12 × 3 mm. Four cell culture chambers for culturing four cells were printed on both ends of the top and bottom. The size of the cylinder above the corner (cell culture or harvest hole) of the square shape was set to 3 × 3 × 3 mm. The printing speed was optimized to a feed rate of 7 mm/s. The printing time for the PLA mold fabrication was approximately 2 h. Ten milliliters of PDMS solution was added to the PLA mold and stored at 25 °C for 16 h. The hardened PMCCC was detached from the petri dish and the PLA mold was removed from PMCCC using a precision tweezer. After, the PMCCC was washed the phosphate-buffered saline solution. The PMCCC was then reattached to the 100 mm cell culture dish printed with PCL 3D scaffolds using the PDMS solution (Figure 1D and Figure 2A(b)).

### 3.2. Cell Culture in the PDMS Multi-Channel Cell Chip with a 3D Scaffold

BNL-CL2 (liver cells), L-132 (lung cells), U87 (human glioblastoma cells), and HeLa cells (human cervical carcinoma) were obtained from the American Type Culture Collection (ATCC, Manassas, VA, USA). The cells were cultured in Dulbecco’s Modified Eagle’s Medium (WELGENE, Gyeongsan-si, Republic of Korea), supplemented with 10% fetal bovine serum (Gibco, Grand Island, NY, USA) and 1% penicillin (WELGENE, Gyeongsan-si, Korea), in a humidified chamber with 5% CO_2_ at 37 °C. The cells (seeding density of 1 × 10^5^ cells/mL) were cultured in the PMCCC, with cell suspension volumes of 430 µL through the cell culture or harvest hole, respectively. After 4 h of culture, the medium was suctioned, and the central hole was filled with fresh medium. After that the cells incubated for 20 h at 37 °C, and then the cells were imaged using an optical microscope (Eclipse TS100; Nikon, Tokyo, Japan).

### 3.3. Morphology of Cells Cultured on Multi-Channel Cell Chip with 3D Scaffolds

The surface morphologies of the cells (1 × 10^5^ cells/mL) cultured on the 2D plate and PMCCC, were analyzed by SEM and optical microscopy. The spheroidal cells were quantified using the SEM image.

### 3.4. Proliferation of Normal and Cancer Cells in the Presence of Anticancer Drugs in the PDMS Multi-Channel Cell Chip

The cells (BNL-CL2, L-132, U87, and HeLa) were seeded in the PMCCC at a density of 1 × 10^5^ cells/mL and incubated for 24 h to allow attachment. After OA (0, 50, 100, 200, and 300 μg/mL in dimethyl sulfoxide (DMSO)) treatment, the cells were incubated for 24 h at 37 °C. Absorbance was measured at 540 nm using a microplate reader (Asys UVM 340; Biochrom Ltd., Cambridge, UK) after the addition of an MTT solution (5 mg/mL) and incubated for 4 h. The proportion of surviving cells was calculated by dividing the average absorbance of the treated cells by that of the untreated cells.

### 3.5. Drug Screening of Normal and Cancer Cells in the PDMS Multi-Channel Cell Chip

Cells were seeded in the PMCCC at a density of 1 × 10^5^ cells/mL and incubated for 24 h at 37 °C to allow attachment. After OA (0, 50, 100, 200, and 300 μg/mL in DMSO) treatment, the cells were incubated for 24 h at 37 °C. To measure LDH release, the culture supernatant (100 μL/well) was transferred to the corresponding well of an optically clear 96-well microplate and analyzed using an LDH cytotoxicity detection kit (Takara Bio, Inc., Otsu, Japan). Absorbance was measured at 490 nm using a microplate reader. Cytotoxicity was calculated by dividing the average absorbance of the treated cells by that of the untreated cells.

### 3.6. Measurement of Apoptosis of Normal and Cancer Cells in the PDMS Multi-Channel Cell Chip

The cells (BNL-CL2, L-132, U87, and HeLa) were seeded in the PMCCC at a density of 1 × 10^5^ cells/mL and incubated for 24 h at 37 °C to allow attachment. After OA (0, 50, 100 μg/mL in DMSO) treatment, the cells were incubated for 24 h at 37 °C. The cells were detached by trypsin and added the medium onto each chamber through the cell culture or harvest hole. After, the cells were harvested, the medium was removed through the cell harvest hole. Apoptosis of the four types of cells cultured on the PMCCC was quantified using an Annexin V-FITC/PI apoptosis detection kit (BD Biosciences, San Diego, CA, USA), according to the manufacturer’s instructions.

### 3.7. Statistical Analysis

Statistical analysis was performed using SPSS (version 18.0; SPSS Inc., Chicago, IL, USA). Averages and standard deviations were calculated and differences between groups were assessed using the analysis of variance method and Duncan’s multiple range test, and differences were considered significant if *p* < 0.05. The correlation of three assays (MTT, LDH, Annexin V-FITC/PI) analyzed the cell viability by Pearson correlation. Cell viability of LDH assay was transferred from cell cytotoxicity data. Moreover, cell viability by Annexin V-FITC/PI was analyzed using the viable cell.

## 4. Conclusions

In this study, we introduced a PMCCC with a PCL 3D scaffolds. These two cell chips could be used for the simultaneous culture of four cell types (BNL-CL2, L-132, U87, and HeLa cells). The surface area at which the cells touched the 3D scaffold of the PMCCC was wider, allowing more cells to attach, and for the attached cells to grow and aggregate to form spheroids inside the scaffold. We demonstrated that the cytotoxic effect of OA increased drug resistance in BNL-CL2, L132, U87, and HeLa cell lines cultured on the PMCCC. In the MTT assay, BNL-CL2, L132, U87, and HeLa cells cultured on PMCCC showed cell proliferation of 107.42%, 43.77%, 79.38% and 48.14% in cells treated with 300 µg/mL OA, respectively. Additionally, LDH assay results showed that BNL-CL2 (66%), L-132 (34.32%), U87 (70.76%), and HeLa cells (39.68%) cultured on the PMCCC had a lower cytotoxicity compared with the 2D plate at 300 µg/mL OA. Annexin V-FITC/PI staining results showed that the PMCCC can be used for simultaneous analysis of the mechanism of cell death, such as necrosis or apoptosis, simultaneously in four different types of cells. Furthermore, we demonstrated correlations between the MTT assay, LDH assay, and Annexin V-FITC/PI. Therefore, we suggest that the PMCCC could be a useful tool for drug screening through a single point injection of multiple cells.

## Figures and Tables

**Figure 1 ijms-22-06997-f001:**
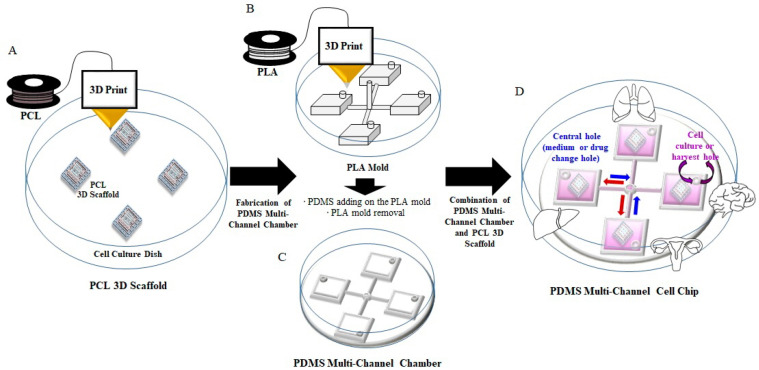
Schematic diagram of the polydimethylsiloxane (PDMS) multi-channel cell chip (PMCCC). (**A**) PCL 3D scaffold. (**B**) PLA mold. (**C**) PDMS multi-channel chamber. (**D**) PMCCC.

**Figure 2 ijms-22-06997-f002:**
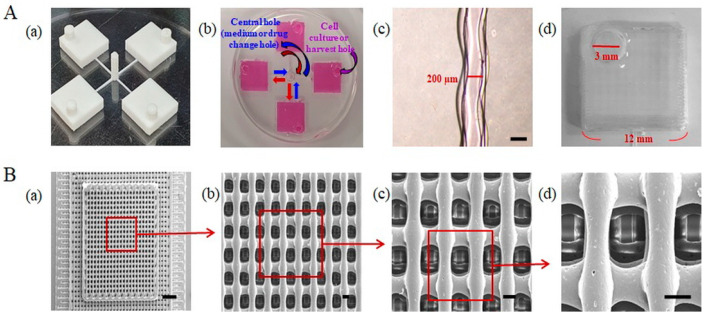
(**A**) Photograph and optical image of polycaprolactone (PCL) mold (**a**), polydimethylsiloxane (PDMS) multi-channel chamber (**b**), each channel (**c**), scale bar: 200 µm) and chamber (**d**). (**B**) Scanning electron microscope (SEM) images of the polycaprolactone (PCL) three-dimensional (3D) scaffold within PDMS multi-channel cell chip chip (**a**, scale bar: 1 mm). Partially magnified SEM image of the 3D scaffold (**b**–**d**, scale bar: 100 µm).

**Figure 3 ijms-22-06997-f003:**
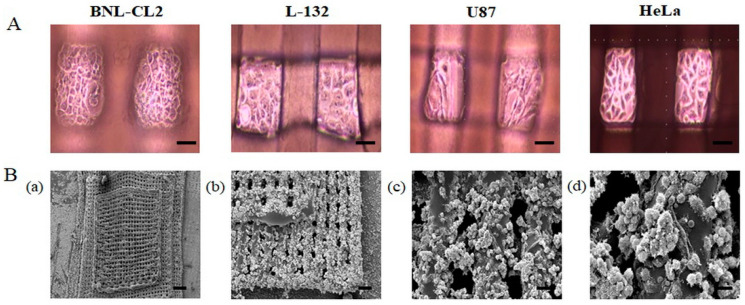
(**A**) Optical images of the four cell types (BNL-CL2, L-132, U87, and HeLa) cultured on a polydimethylsiloxane (PDMS) multi-channel cell chip with a polycaprolactone (PCL) three-dimensional (3D) scaffold. (**B**) SEM image of HeLa cells cultured on the PDMS multi-channel cell chip with a PCL 3D scaffold (**a**); scale bar: 1 mm). Partially magnified SEM image of HeLa cells cultured on the first and second floors of the 3D structure (**b**); scale bar: 200 µm). Partially magnified SEM image of HeLa cells cultured on the first floor of the 3D structure (**c**); scale bar: 50 µm). Partially magnified SEM image of the third floor of the 3D structure (**d**); scale bar: 10 µm).

**Figure 4 ijms-22-06997-f004:**
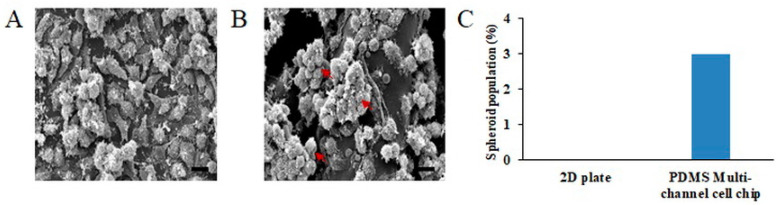
Scanning electron microscope (SEM) images of the comparison of HeLa cell morphology in a two-dimensional (2D) plate (**A**) and, polydimethylsiloxane (PDMS) multi-channel cell chip (**B**) (scale bar: 10 µm). Results of the number of spheroid cells obtained by SEM image (**C**).

**Figure 5 ijms-22-06997-f005:**
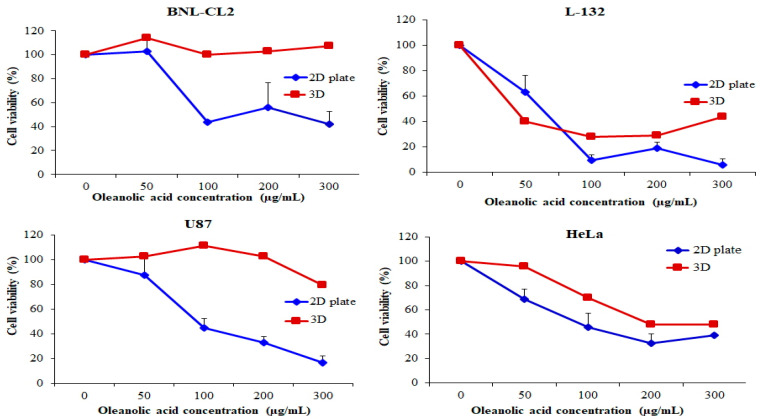
Cell viability evaluation of normal cell lines (BNL-CL2 and L-132) and cancer cell lines (U87 and HeLa) cultured in a two-dimensional (2D) plate and polydimethylsiloxane (PDMS) multi-channel cell chip, using the 3-(4,5-dimethylthiazol-2-yl)-2,5-diphenyl tetrazolium bromide (MTT) assay. The cells were cultured at 1 × 10^5^ cells/mL and treated with oleanolic acid (OA; 50, 100, 200, 300 µg/mL) for 24 h. The relative cell viability was calculated as the percentage of untreated cells. The data are represented as mean ± SD, *n* = 3.

**Figure 6 ijms-22-06997-f006:**
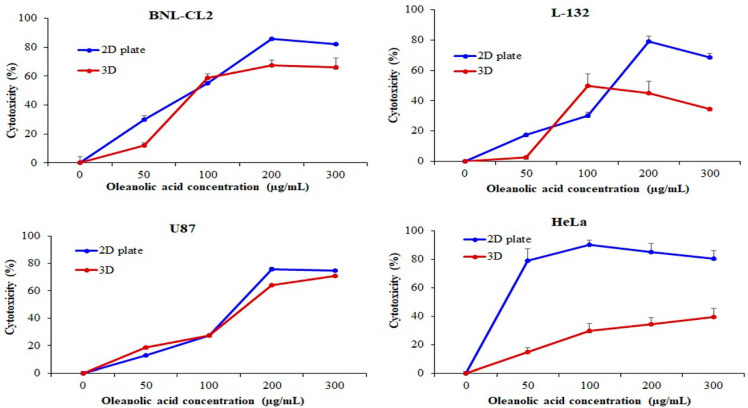
Cytotoxicity evaluation of normal cell lines (BNL-CL2 and L-132) and cancer cell lines (U87 and HeLa) cultured in a two-dimensional (2D) plate and polydimethylsiloxane (PDMS) multi-channel cell chip using the lactate dehydrogenase (LDH) assay. The cells were cultured at 1 × 10^5^ cells/mL and treated with oleanolic acid (OA; 50, 100, 200, 300 µg/mL) for 24 h. The relative cell viability was calculated as the percentage of untreated cells. The data are represented as mean ± SD, *n* = 3.

**Figure 7 ijms-22-06997-f007:**
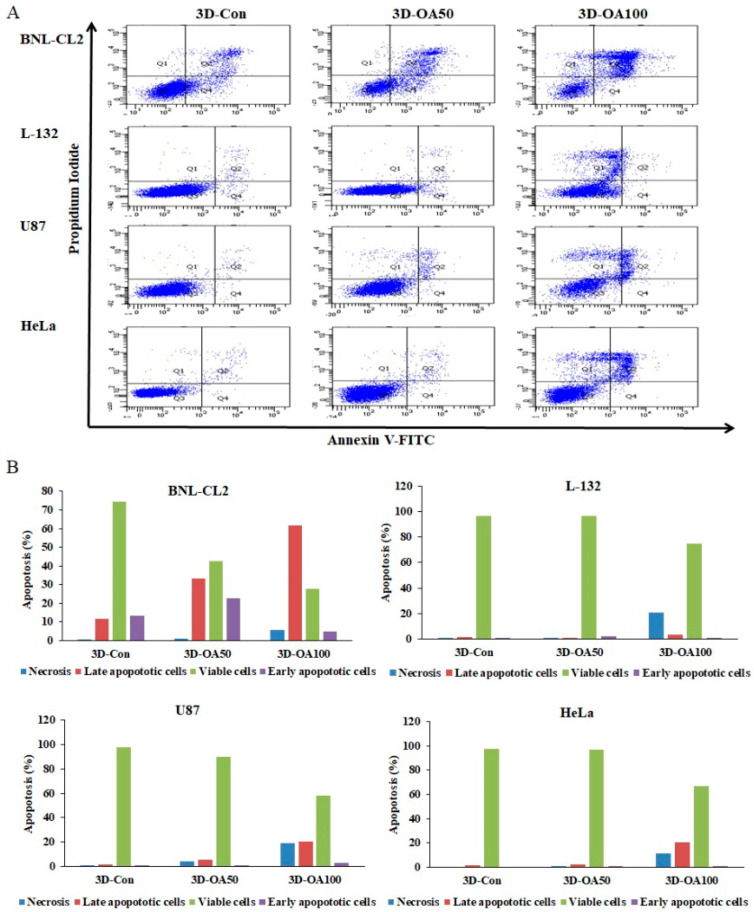
(**A**) Determination of the apoptotic populations of normal cells (BNL-CL2 and L-132 cells) and cancer cells (U87 and HeLa cells) Annexin V-fluorescein isothiocyanate (FITC)/propidium iodide (PI) staining and subjecting to flow cytometry in polydimethylsiloxane (PDMS) multi-channel cell chip. The cells were cultured at 1 × 10^5^ cells/mL and treated with oleanolic acid (OA; 50 and 100 µg/mL) for 24 h. (**B**) Quantitation of the data shown in (**A**).

**Figure 8 ijms-22-06997-f008:**
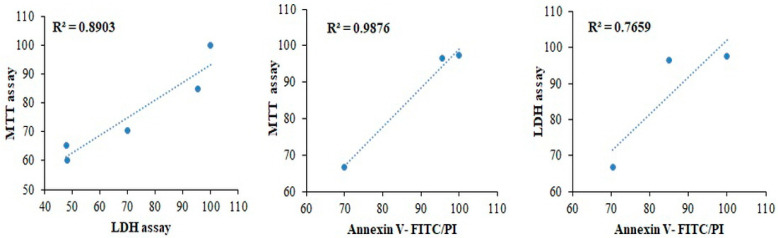
Correlation of 3-(4,5-dimethylthiazol-2-yl)-2,5-diphenyl tetrazolium bromide (MTT) assay, lactate dehydrogenase (LDH) assay and Annexin V-fluorescein isothiocyanate (FITC)/propidium iodide (PI) staining of drug treatment on polydimethylsiloxane (PDMS) multi-channel cell chip cultured HeLa cells.

**Table 1 ijms-22-06997-t001:** Printer settings employed for polycaprolactone (PCL) 3D scaffold and polylactic acid (PLA) mold.

Parameter	PCL 3D Scaffold	PLA Mold
Nozzle diameter	0.2 mm	0.2 mm
Layer height	0.1 mm	0.1 mm
Fill density	40%	100%
Fill pattern	Lines	Lines
Extruder temperature	85 ℃	210 ℃
Bed temperature	5 ℃	25 ℃
Speed of print moves	2 mm/s	7 mm/s

## Data Availability

Not applicable.

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
