# Peer review of "Drug Evaluation Based on a Multi-Channel Cell Chip with a Horizontal Co-Culture"

_ijms, 2021, doi:10.3390/ijms22136997_

Round 1

Reviewer 1 Report

The manuscript ijms-1237112 entitled " Drug Evaluation based on Multi-channel Cell Chip to Depict Human Tissue " presents the PDMS multi-channel cell chip with a PCL 3D scaffolds. The proposed chip seems to be a very good idea with a wide range of applications. As the authors declare: “Additionally, the PDMS multi-channel cell chip allowed simultaneous observation of the effects of one drug on four different cell types” which has been proven. In addition to the fabrication of multi-channel cell chip containing a three-dimensional (3D) scaffold, authors conducted number of studies confirming its usefulness.

Horizontal co-culture and drug toxicity screening in multi-organ culture (human glioblastoma, cervical cancer, normal liver cells, and normal lung cells) was successfully performed. Authors investigated cell proliferation and cytotoxicity by conducting assays of 3-(4,5-dimethylthiazol-2-yl)-2,5-18 dphenyltetrazolium bromide (MTT) and lactate dehydrogenase (LDH) as well as analysis of oleanolic acid (OA)-treated multi-channel cell chips. The results of the MTT and LDH assays showed that OA treatment in the multi-channel cell chip of four cell lines enhanced chemoresistance of cells compared with that in the 2D culture. Furthermore, they demonstrated the feasibility of the application of the chip in various analysis methods through Annexin V–fluorescein isothiocyanate/propidium iodide staining, which is not used for conventional cell chips.

This is an interesting work and the results and discussions are well presented. I suggest to accept the manuscript with several corrections:

- For me conclusion is too short for that wide article. I propose to list the most important achievements.

- The photo (d) in section A of Figure 2 is of poor quality. I recommend changing for the better resolution / at a different angle.

-Figure 4 C.  The authors interpret as: “Furthermore, the PDMS multi-channel cell chip showed more aggregated cells than the 2D cell culture”.  Since no aggregation is visible for 2D culture. It cannot be interpreted that way.

- I am not sure if the title fully fits the manuscript theme.

- The name “PDMS multi-channel cell chip” is used 71 times. I would suggest to abbreviate the name (for example PMCCC) to make the text more concise

- The chip fabrication steps should be clearly listed as the description in section 3 is not entirely clear.

- instead of photos a simple and clear schematic drawing of the chip final structure would help to understand its construction.

- please describe how the separation between “floors” of the scaffold is achieved and clearly state the number of “floors” in each scaffold.

- lines 115 -119. Please rewrite this section. It is not clear if the evaporation form the open structure of the chip is acceptable.

- lines 335 -336. There are dimensions for the first and second “floor”. Please add dimensions for the third “floor”.

- section 3.2 the  information how the cell suspension was introduced into the chip is not clear. One might assume that the “harvest hole” is used for that but line 368 also mentions “cell injection hole”. Please correct this section.

- is there any chance of migration of the cell suspension form one chip to another? Please address this issue.

- line 365. Please check the penicillin concentration unit.

- section 3.2. Please state the incubation time.

- section 3.4; 3.5; 3.6. Please state the temperature of the initial incubation.

Author Response

Response Letter to Reviewer’s Comments

Manuscript ID: ijms-1237112

Title: Drug Evaluation based on Multi-Channel Cell Chip with Hor-izontal Co-Culture

Reviewer 1

The manuscript ijms-1237112 entitled " Drug Evaluation based on Multi-channel Cell Chip to Depict Human Tissue " presents the PDMS multi-channel cell chip with a PCL 3D scaffolds. The proposed chip seems to be a very good idea with a wide range of applications. As the authors declare: “Additionally, the PDMS multi-channel cell chip allowed simultaneous observation of the effects of one drug on four different cell types” which has been proven. In addition to the fabrication of multi-channel cell chip containing a three-dimensional (3D) scaffold, authors conducted number of studies confirming its usefulness.

Horizontal co-culture and drug toxicity screening in multi-organ culture (human glioblastoma, cervical cancer, normal liver cells, and normal lung cells) was successfully performed. Authors investigated cell proliferation and cytotoxicity by conducting assays of 3-(4,5-dimethylthiazol-2-yl)-2,5-18 dphenyltetrazolium bromide (MTT) and lactate dehydrogenase (LDH) as well as analysis of oleanolic acid (OA)-treated multi-channel cell chips. The results of the MTT and LDH assays showed that OA treatment in the multi-channel cell chip of four cell lines enhanced chemoresistance of cells compared with that in the 2D culture. Furthermore, they demonstrated the feasibility of the application of the chip in various analysis methods through Annexin V–fluorescein isothiocyanate/propidium iodide staining, which is not used for conventional cell chips.

This is an interesting work and the results and discussions are well presented. I suggest to accept the manuscript with several corrections:

  • For me conclusion is too short for that wide article. I propose to list the most important achievements.Page 12
  1. à Thank you for your valuable comments. We have revised the conclusion.

In this study, we introduced a PMCCC with a PCL 3D scaffolds. These two cell chips could be used for the simultaneous culture of four cell types (BNL-CL2, L-132, U87, and HeLa cells). The surface area at which the cell touched the 3D scaffold of the PMCCC was wider, allowing more cells to attach, and the attached cells grow and aggregate to form spheroids inside the scaffold. We demonstrated that the cytotoxic effect of OA increased drug resistance in BNL-CL2, L132, U87, and HeLa cell lines cultured on the PMCCC. In MTT assay, BNL-CL2, L132, U87, and HeLa cells cultured on PMCCC showed cell proliferation of 107.42%, 43.77%, 79.38% and 48.14% in cell treated with 300 µg/mL OA, respectively. Additionally, LDH assay results showed that BNL-CL2 (66%), L-132 (34.32%), U87(70.76%), and HeLa cells (39.68%) cultured on the PMCCC have lower cytotoxicity compared with 2D plate at 300 µg/mL OA. Annexin V-FITC/PI staining results showed that the PMCCC can be used for simultaneous analysis of the mechanism of cell death, such as necrosis or apoptosis, simultaneously in four different types of cells. Furthermore, we demonstrated correlations between the MTT assay, LDH assay, and Annexin V-FITC/PI. Therefore, we suggest that the PMCCC could be a useful tool for drug screening through a single point injection of multiple cells.

  • The photo (d) in section A of Figure 2 is of poor quality. I recommend changing for the better resolution at a different angle. Page 3
    1. à We have revised the Figure 2A(d) image.
  • Figure 4 C.  The authors interpret as: “Furthermore, the PDMS multi-channel cell chip showed more aggregated cells than the 2D cell culture”.  Since no aggregation is visible for 2D culture. It cannot be interpreted that way.Page 5
  1. à We have revised the sentence of Figure 4C.

2.4. The Comparison of Cell Morphology of the PDMS Multi-channel Cell Chips and 2D Cell Culture

In Figures 4, we compared the cell morphology in the same area of the 2D plate and PMCCC. The HeLa cell culture on the 2D plate showed a flat morphology in Figure 4A. Whereas, seeded cells on the PCL 3D scaffolds were infiltrated, attached and proliferated inside the scaffolds Figure 4B. Additionally, HeLa cells cultured on the PMCCC were partially flat or displayed an aggregated morphology. Furthermore, the surface of cells cultured in the PMCCC secreted more extracellular matrix compared with the surface of 2D cultured cells. In addition, Figure 4C showed the number of aggregation cell in PMCCC and 2D cell culture. Three aggregated cells were observed on a PMCCC. However, 2D cell culture has not shown aggregation cells. The interconnected pores of 3D scaffolds facilitate the transport of biomolecules and cells. [26]. Therefore, we performed drug screening on the PMCCC combined with PCL 3D scaffolds compare to 2D cultured cells. Additionally, we focused on whether conventional molecular biology analysis methods, such as MTT LDH assays, and fluorescence-activated cell sorting, can be applied to the PMCCC.

  • I am not sure if the title fully fits the manuscript theme.

à We have revised the title.

Page 1

Drug Evaluation based on Multi-channel Cell Chip with Horizontal Co-culture

  • The name “PDMS multi-channel cell chip” is used 71 times. I would suggest to abbreviate the name (for example PMCCC) to make the text more concise
  1. à We have revised the manuscripts

  • The chip fabrication steps should be clearly listed as the description in section 3 is not entirely clear.

à We have revised the description in section 3.

Page 10-11

We showed the fabrication process of PMCCC in Figure 1. A 3D scaffold consisting of three floors structure was fabricated by FDM technology using PCL at 85 °C on a cell culture dish (Table 1). A 3D bioprinter (INVIVO; ROKIT, Geumcheon-gu, Seoul, Korea) was set such that PCL was extruded through a nozzle (diameter: 0.2 mm) and deposited in accordance with predetermined designs. The NewCreatorK program (Version 1.57.70, ROKIT INVIVO Corp., Seoul, Republic of Korea) was used to generate the G-code to convert the 3D object files for compatibility with a 3D bioprinter. The printing speed was optimized to a feed rate of 2 mm/s. The sizes of the first, second and third floors were set to 6 × 6 × 0.25 mm, 8 × 8 × 0.25 mm and 10 × 10 × 0.25 mm, respectively. The distance between each of the three floors was set to 0.2 mm. The PCL 3D scaffolds was fabricated four on the cell dish. The printing time for the three-floor structure was approximately 2 hr. We printed four PCL scaffolds on a 100 mm cell dish as shown in Figure 1A. The morphology of the PCL 3D scaffold was visualized using a scanning electron microscope (SEM) (ZEISS GEMINI2, Oberkochen, Germany).

The multi-channel cell chip was fabricated by a 3D bioprinter, using PLA filaments and PDMS as shown in Figure 1B and C. The PLA filaments were extruded through a nozzle (diameter, 0.2 mm) and deposited in accordance with predetermined designs using the NewCreatorK program on the 60 mm cell culture dish (Table 1). The PLA mold showed in Figure 2A(a). The horizontal and vertical line sizes size was set to 24 × 0.3 × 0.3 mm. The size of the cylinder in the middle (central hole: medium change hole) of the line was 2 × 2 × 5 mm. The size of the square structure to make each cell culture chamber was designed to be 12 × 12 × 3mm. Four cell culture chambers for culturing four cells were printed on both ends of the top and bottom. The size of the cylinder above the corner (cell culture or harvest hole) of the square shape was set to 3 × 3 × 3 mm. The printing speed was optimized to a feed rate of 7 mm/s. The printing time for the PLA mold fabrication was approximately 2 h. Ten milliliters of PDMS solution was added to the PLA mold and stored at 25°C for 16 h. The hardened PMCCC was detached from the petri dish and the PLA mold was removed from PMCCC using a precision tweezer. After, the PMCCC was washed the PBS. The PMCCC was then reattached to the 100 mm cell culture dish printed with PCL 3D scaffolds using the PDMS solution (Figure 1D and 2A(b)).

  • instead of photos a simple and clear schematic drawing of the chip final structure would help to understand its construction.

à We have presented the schematic drawing such as Figure 1D.

 Page 3

  • please describe how the separation between “floors” of the scaffold is achieved and clearly state the number of “floors” in each scaffold.

à We have not detached the PCL scaffold. We described the section 3.

à Also, we have added the number of floors information.

Page 10

We showed the fabrication process of PMCCC in Figure 1. A 3D scaffold consisting of three floors structure was fabricated by FDM technology using PCL at 85 °C on a cell culture dish (Table 1). A 3D bioprinter (INVIVO; ROKIT, Geumcheon-gu, Seoul, Korea) was set such that PCL was extruded through a nozzle (diameter: 0.2 mm) and deposited in accordance with predetermined designs. The NewCreatorK program (Version 1.57.70, ROKIT INVIVO Corp., Seoul, Republic of Korea) was used to generate the G-code to convert the 3D object files for compatibility with a 3D bioprinter. The printing speed was optimized to a feed rate of 2 mm/s. The sizes of the first, second and third floors were set to 6 × 6 × 0.25 mm, 8 × 8 × 0.25 mm and 10 × 10 × 0.25 mm, respectively. The distance between each of the three floors was set to 0.2 mm. The PCL 3D scaffolds was fabricated four on the cell dish. The printing time for the three-floor structure was approximately 2 hr. We printed four PCL scaffolds on a 100 mm cell dish as shown in Figure 1A. The morphology of the PCL 3D scaffold was visualized using a scanning electron microscope (SEM) (ZEISS GEMINI2, Oberkochen, Germany).

  • lines 115 -119. Please rewrite this section. It is not clear if the evaporation form the open structure of the chip is acceptable.

à We have revised the sentence.

Page 3-4

We observed the possibility of culture medium migration in a multi-channel cell chip. As shown in Figure 2A(c), the channel width of the PMCCC was 200 µm. The PMCCC showed simultaneous migration of the culture medium. Generally, the width of a channel directly affects fluid flow [15]. To maintain laminar flow within the chip, the channels should be as small as required to maintain the functionality of the chip [15]. The diameter of a cell ranges from 5 μm to 50 μm [15]. For cells to have a good cell–cell interaction, multiple cells must be present together [15]. This means that the minimum channel width should allow at least four cells to attach perpendicularly (side by side) to the fluid flow [15]. Hence, the minimum width should be 200 μm [16]. Goh et al. suggested an affordable tool for the fabrication of microchannels measuring 200 µm in width and 100 µm in height, using polyvinyl alcohol, and demonstrated that they are compatible with a wide range of polymeric matrixes [17]. Saggiomo et al. used an FDM technique to extrude an inexpensive and commercially available acrylonitrile butadiene styrene (ABS) scaffold using a 500-µm diameter nozzle. The scaffold was then cured with PDMS and acetone was added to dissolve the ABS scaffold to fabricate a microfluidic device [18]. Our PMCCC was fabricated using industrial polylactic acid (PLA) filaments, with a channel width of 200 µm. As shown in Figure 2A(d), the cell culture chamber size of the PMCCC and silicon wafer multi-channel cell chip were 12 × 12 × 3 mm and 5 × 5 × 0.5 mm, respectively. The PMCCCs could hold media volumes of 430 µL. Evaporation can negatively affect the cell culture process because of high osmotic pressure and concentrating effects [19-21]. Our PMCCC has low effect from evaporation. Therefore, the PMCCC was suitable for the cell culture.

  • lines 335 -336. There are dimensions for the first and second “floor”. Please add dimensions for the third “floor”.

à We have revised the floor size.

Page 10

3.1. Fabrication of a PDMS Multi-channel Cell Chip with a 3D Scaffold using a 3D Bioprinter

We showed the fabrication process of PMCCC in Figure 1. A 3D scaffold consisting of three floors structure was fabricated by FDM technology using PCL at 85 °C on a cell culture dish (Table 1). A 3D bioprinter (INVIVO; ROKIT, Geumcheon-gu, Seoul, Korea) was set such that PCL was extruded through a nozzle (diameter: 0.2 mm) and deposited in accordance with predetermined designs. The NewCreatorK program (Version 1.57.70, ROKIT INVIVO Corp., Seoul, Republic of Korea) was used to generate the G-code to convert the 3D object files for compatibility with a 3D bioprinter. The printing speed was optimized to a feed rate of 2 mm/s. The sizes of the first, second and third floors were set to 6 × 6 × 0.25 mm, 8 × 8 × 0.25 mm and 10 × 10 × 0.25 mm, respectively. The distance between each of the three floors was set to 0.2 mm. The PCL 3D scaffolds was fabricated four on the cell dish. The printing time for the three-floor structure was approximately 2 hr. We printed four PCL scaffolds on a 100 mm cell dish as shown in Figure 1A. The morphology of the PCL 3D scaffold was visualized using a scanning electron microscope (SEM) (ZEISS GEMINI2, Oberkochen, Germany).

  • section 3.2 the information how the cell suspension was introduced into the chip is not clear. One might assume that the “harvest hole” is used for that but line 368 also mentions “cell injection hole”. Please correct this section.

à Our harvest hole same with the cell culture hole. We revised the “cell injection hole” to “cell culture or harvest hole”

Page 11

The cells (seeding density of 1 × 105 cells/mL) were cultured in the PMCCC, with cell suspension volumes of 430 µL trough the cell culture or harvest hole, respectively.

  • is there any chance of migration of the cell suspension form one chip to another? Please address this issue.

à Our cell chip has not shown migration of cell suspension. To prove this, we put in the four media of different colors and checked it as shown in the photo.

  1. line 365. Please check the penicillin concentration unit.

à we revised the penicillin concentration unit

Page 11

The cells were cultured in Dulbecco’s modified Eagle’s medium (Welgene, Gyeongsan-si, Republic of Korea), supplemented with 10% fetal bovine serum (Gibco, Grand Island, NY, USA) and 1% penicillin (Welgene, Gyeongsan-si, Republic of Korea), in a humidified chamber with 5% CO2 at 37 °C.

  1. section 3.2. Please state the incubation time.

à We added the incubation time.

Page 11

After 4 h of culture, the medium was suctioned, and the central hole was filled with fresh medium. After that the cells incubated for 20h at 37 °C, and then the cells were imaged using an optical microscope.

  1. section 3.4; 3.5; 3.6. Please state the temperature of the initial incubation.

à We added the temperature of the initial incubation in section 3.4; 3.5; 3.6.

Page 11-12

3.4. Proliferation of Normal and Cancer Cells in the Presence of Anticancer Drugs in the PDMS Multi-Channel Cell Chip

The cells (BNL-CL2, L-132, U87, and HeLa) were seeded in the PMCCC at a density of 1×105 cells/mL and incubated for 24 h at 37 °C to allow attachment. After OA (0, 50, 100, 200, and 300 μg/mL in dimethyl sufoxide (DMSO)) treatment, the cells were incubated for 24 h at 37 °C. Absorbance was measured at 540 nm using a microplate reader (Asys UVM 340; Biochrom Ltd., Cambridge, UK) after the addition of MTT solution (5 mg/mL) and incubation for 4 h. The proportion of surviving cells was calculated by dividing the average absorbance of the treated cells by that of the untreated cells.

3.5. Drug Screening of Normal and Cancer Cells in the PDMS Multi-Channel Cell Chip

Cells were seeded in the PMCCC at a density of 1×105 cells/mL and incubated for 24 h at 37 °C to allow attachment. After OA (0, 50, 100, 200, and 300 μg/mL in DMSO) treatment, the cells were incubated for 24 h at 37 °C. To measure LDH release, the culture supernatant (100 μL/well) was transferred to the corresponding well of an optically clear 96 well microplate and analyzed using an LDH cytotoxicity detection kit (Takara Bio, Inc., Otsu, Japan). Absorbance was measured at 490 nm using a microplate reader (Asys UVM 340; Biochrom Ltd., Cambridge, UK). Cytotoxicity was calculated by dividing the average absorbance of the treated cells by that of the untreated cells.

3.6. Measurement of Apoptosis of Normal and Cancer Cells in the PDMS Multi-channel Cell Chip

The cells (BNL-CL2, L-132, U87, and HeLa) were seeded in the PMCCC at a density of 1×105 cells/mL and incubated for 24 h at 37 °C to allow attachment. After OA (0, 50, 100 μg/mL in DMSO) treatment, the cells were incubated for 24 h at 37 °C. The cells were detached by trypsin and added the medium on each chamber through cell injection hole. After, the cells were harvested the medium the through cell culture or harvest hole. Apoptosis of the four types of cells cultured on the PMCCC was quantified using an Annexin V-FITC apoptosis detection kit (BD Biosciences, San Diego, CA, USA), according to the manufacturer’s instructions.

Reviewer 2 Report

The authors describe the development of a Multi-channel Cell Chip and examine the proliferation and toxicity of human cancer and normal cells in the presence of oleanolic acid. The paper is allover well written.

I have a couple of minor points that may help to further improve the manuscript.

In figure 2 the scale bar is not very clearly depicted. Please add the information about the scale bare in figure caption as is presented in Figure 3 and 4

Could you please add in the experimental section ‘’3.1. Fabrication of a PDMS Multi-channel Cell Chip with a 3D Scaffold using a 3D Bioprinter” a description of how the PLA mold was removed from the PDMS multi-channel cell chip.

Author Response

Response Letter to Reviewer’s Comments

Manuscript ID: ijms-1237112

Title: Drug Evaluation based on Multi-Channel Cell Chip with Hor-izontal Co-Culture

Reviewer 2

The authors describe the development of a Multi-channel Cell Chip and examine the proliferation and toxicity of human cancer and normal cells in the presence of oleanolic acid. The paper is allover well written.

I have a couple of minor points that may help to further improve the manuscript.

  • In figure 2 the scale bar is not very clearly depicted. Please add the information about the scale bare in figure caption as is presented in Figure 3 and 4

à Thank you for your valuable comments. We added the scale bar information in figure 2 caption.

Page 3

 Figure 2. (A) Photograph and optical image of polycaprolactone (PCL) mold (a), polydimethylsiloxane (PDMS) multi-channel chamber (b), each channel (c, scale bar: 200 µm) and chamber (d). (B) Scanning electron microscopic (SEM) images of the polycaprolactone (PCL) three-dimensional (3D) scaffold within PDMS multi-channel cell chip (a-d).

  • Could you please add in the experimental section ‘’3.1. Fabrication of a PDMS Multi-channel Cell Chip with a 3D Scaffold using a 3D Bioprinter” a description of how the PLA mold was removed from the PDMS multi-channel cell chip.

à We detached the PLA mold from PMCC using a precision tweezer. We revised the method section.

Page 10

3.1. Fabrication of a PDMS Multi-channel Cell Chip with a 3D Scaffold using a 3D Bioprinter

We showed the fabrication process of PMCCC in Figure 1. A 3D scaffold consisting of three floors structure was fabricated by FDM technology using PCL at 85 °C on a cell culture dish (Table 1). A 3D bioprinter (INVIVO; ROKIT, Geumcheon-gu, Seoul, Korea) was set such that PCL was extruded through a nozzle (diameter: 0.2 mm) and deposited in accordance with predetermined designs. The NewCreatorK program (Version 1.57.70, ROKIT INVIVO Corp., Seoul, Republic of Korea) was used to generate the G-code to convert the 3D object files for compatibility with a 3D bioprinter. The printing speed was optimized to a feed rate of 2 mm/s. The sizes of the first, second and third floors were set to 6 × 6 × 0.25 mm, 8 × 8 × 0.25 mm and 10 × 10 × 0.25 mm, respectively. The distance between each of the three floors was set to 0.2 mm. The PCL 3D scaffolds was fabricated four on the cell dish. The printing time for the three-floor structure was approximately 2 hr. We printed four PCL scaffolds on a 100 mm cell dish as shown in Figure 1A. The morphology of the PCL 3D scaffold was visualized using a scanning electron microscope (SEM) (ZEISS GEMINI2, Oberkochen, Germany).

The multi-channel cell chip was fabricated by a 3D bioprinter, using PLA filaments and PDMS as shown in Figure 1B and C. The PLA filaments were extruded through a nozzle (diameter, 0.2 mm) and deposited in accordance with predetermined designs using the NewCreatorK program on the 60 mm cell culture dish (Table 1). The PLA mold showed in Figure 2A(a). The horizontal and vertical line sizes size was set to 24 × 0.3 × 0.3 mm. The size of the cylinder in the middle (central hole: medium change hole) of the line was 2 × 2 × 5 mm. The size of the square structure to make each cell culture chamber was designed to be 12 × 12 × 3mm. Four cell culture chambers for culturing four cells were printed on both ends of the top and bottom. The size of the cylinder above the corner (cell culture or harvest hole) of the square shape was set to 3 × 3 × 3 mm. The printing speed was optimized to a feed rate of 7 mm/s. The printing time for the PLA mold fabrication was approximately 2 h. Ten milliliters of PDMS solution was added to the PLA mold and stored at 25°C for 16 h. The hardened PMCCC was detached from the petri dish and the PLA mold was removed from PMCCC using a precision tweezer. After, the PMCCC was washed the PBS. The PMCCC was then reattached to the 100 mm cell culture dish printed with PCL 3D scaffolds using the PDMS solution (Figure 1D and 2A(b)).
